# The Spectrum of Still’s Disease: A Comparative Analysis of Phenotypic Forms in a Cohort of 238 Patients

**DOI:** 10.3390/jcm11226703

**Published:** 2022-11-12

**Authors:** Pierre-Antoine Neau, Thomas El-Jammal, Clément Javaux, Nicolas Fournier, Orlane Chol, Léopold Adelaïde, Kim Heang Ly, Mathieu Gerfaud-Valentin, Laurent Perard, Marine Fouillet-Desjonqueres, Julie Le Scanff, Emmanuelle Vignot, Arnaud Hot, Alexandre Belot, Isabelle Durieu, Pascal Sève, Yvan Jamilloux

**Affiliations:** 1Department of Internal Medicine, Croix-Rousse University Hospital, Hospices Civils de Lyon, Université Claude Bernard-Lyon 1, 69004 Lyon, France; 2Department of Internal Medicine, Grenoble Alpes University Hospital, 38043 Grenoble, France; 3Department of Internal Medicine, Vienne-Lucien HUSSEL Hospital, 38200 Vienne, France; 4Department of Internal Medicine, Limoges University Hospital, 87042 Limoges, France; 5Department of Internal Medicine, Saint Luc Saint Joseph Hospital, 69007 Lyon, France; 6Department of Pediatric Nephrology, Rheumatology, Dermatology, Mère-Enfant Hospital, Hospices Civils de Lyon, Université Claude Bernard-Lyon 1, 69500 Bron, France; 7Department of Internal Medicine, Villefranche-sur-Saône Hospital, 69400 Gleize, France; 8Department of Rheumatology, Edouard Herriot University Hospital, Hospices Civils de Lyon, Université Claude Bernard-Lyon 1, 69003 Lyon, France; 9Department of Internal Medicine, Edouard Herriot University Hospital, Hospices Civils de Lyon, Université Claude Bernard-Lyon 1, 69003 Lyon, France; 10CIRI (Centre International de Recherche en Infectiologie), Inserm U1111, CNRS, UMR5308, ENS de Lyon, Université Claude Bernard Lyon 1, 69007 Lyon, France; 11Lyon Immunopathology Federation (LIFE), 69000 Lyon, France; 12Department of Internal Medicine, Lyon-Sud University Hospital, Hospices Civils de Lyon, Université Claude Bernard-Lyon 1, 69310 Pierre-Benite, France; 13Research on Healthcare Performance (RESHAPE), INSERM U1290, Université Claude Bernard Lyon 1, 69003 Lyon, France

**Keywords:** adult-onset still’s disease, systemic juvenile idiopathic arthritis, autoinflammatory disease, prognosis, arthritis

## Abstract

Still’s disease (SD) is a heterogeneous autoinflammatory disorder for which several phenotypes have been described. We conducted a retrospective study to re-evaluate the dichotomous view of the disease, to compare the juvenile and adult forms, and to look for prognostic factors. We collected data from ten French centers, seeking patients with a diagnosis of adult-onset SD (AOSD) or systemic juvenile idiopathic arthritis (sJIA). We identified 238 patients, 152 (64%) of whom had AOSD while 86 (36%) had sJIA. The median age at SD onset was 26.6 years. In patients with identifiable patterns, the course of SD was systemic in 159 patients (74%), chronic in 55 (26%). Sore throat and myalgia were more frequent in patients with AOSD. Abnormal liver tests, serum ferritin and C-reactive protein levels were higher in AOSD group. Fever and skin rash were predictive of complete remission or recovery and high lactate dehydrogenase level was a poor prognosis factor. Symptoms such as splenomegaly, skin rash, high polymorphonuclear neutrophils count and macrophage activation syndrome were predictive of a systemic phenotype. Overall, there were no major differences between sJIA and AOSD. Our results are consistent with the “biphasic” model of an autoinflammatory disease that can progress to chronic arthritis if not treated early.

## 1. Introduction

Still’s disease (SD) is a rare systemic autoinflammatory disorder of unknown etiology [1]. Adult-onset Still’s disease (AOSD) usually affects young adults, with a mean age of diagnosis generally between the second and third decade [2,3,4], although some recent studies have reported an age of onset around fifty years [5,6]. By definition, the term AOSD applies to patients with disease onset after the age of 16, but evidence largely suggests that AOSD and systemic juvenile idiopathic arthritis (sJIA) form a continuum, within the same disease spectrum [7,8,9]. A definitive terminology for this spectrum of disease has yet to be established but the name “Still’s disease” is currently in common use.

In the absence of a specific clinical or biological marker, the diagnosis of SD is made on the basis of a combination of evidence, after eliminating differential diagnoses. The diagnostic process can thus be lengthy. For convenience, the classification criteria of Yamaguchi [10] and/or Fautrel [11] are often used to make the diagnosis. Consequently, subsequent validation of these sets of criteria, on retrospective series, may suffer from the absence of a gold standard and a circular reasoning bias.

Depending on the course of the disease, three phenotypic subtypes have previously been described: monocyclic, typified by a single flare; polycyclic, characterized by recurrent flares interspersed with complete remission; and chronic, characterized by a non-remittent disease with destructive arthritis [12,13]. Another model, based on cytokine expression and clinicobiological data, proposed a dichotomous view of SD including a systemic phenotype and a chronic articular phenotype [14,15]. In the era of biologics and with the emergence of the concept of a therapeutic window of opportunity, it seems likely that this dichotomy rather illustrates a biphasic evolution, with the chronic articular form occurring in the case of insufficiently controlled disease in its early phase [16,17]. However, several questions still remain as to the best treatment strategy, in particular the importance of corticosteroid therapy in combination with biotherapies (or not) as first-line treatment.

In this study, we sought to assess the heterogeneity of the SD spectrum. We analyzed a multicenter cohort of patients with Still’s disease with three main objectives: (1) to re-evaluate the dichotomous view of the disease; (2) to compare sJIA and AOSD; and (3) to look for predictive factors for progression and outcome.

## 2. Materials and Methods

### 2.1. Identification and Selection of sJIA and AOSD Cases

Patients followed for sJIA or AOSD in ten French hospitals, between January 2001 and June 2022, were included in the observational registry (AURAL STILL study). Patients were identified by an automated search in the electronic patient record software Easily^®^ (HCL-Hospice, Lyon, France) and through a call for observations. The diagnosis of SD was retained if it was confirmed by the referring physician at the last follow-up. All the records were reviewed and validated by at least two independent investigators (P.A.N.; N.F.; C.J. and Y.J.). Conflicting cases were evaluated by a third expert (P.A.N.; N.F.; C.J., Y.J., P.S., or M.G.V.) and, if necessary discussed and classified by consensus. The International League of Associations for Rheumatology (ILAR) [18], Pediatric Rheumatology International Trials Organization (PRINTO) [19], Fautrel’s and Yamaguchi’s classification criteria were systematically sought (see below) to assist in the classification of patients but were not used to retain the diagnosis. Cases with insufficient data (50% or more) or uncertain diagnosis were excluded.

### 2.2. Data Collection

All data were collected retrospectively in an anonymous and standardized electronic Case Report Form (Ennov Clinical, v.7.5.720.1, Ennov, Paris, France), created for the AURAL STILL study.

At the time of diagnosis, demographic data, medical history, clinical and biological data, as well as imaging and histological data were collected. Laboratory parameters included blood cell counts (including white blood cell count, WBC), coagulation parameters, lactate dehydrogenase (LDH), C-reactive protein (CRP), liver function tests, triglyceride concentrations, total serum ferritin (SF) and glycosylated fraction of ferritin (GF, as a % of SF), rheumatoid factor (RF) and antinuclear autoantibodies (ANA). ANA were searched by indirect immunofluorescence, with titers ≥160 were considered positive. Pleuritis/pleurisy was assessed by a double check of the investigator on medical imaging when available, in addition to medical reports. Pericarditis was confirmed by reviewing the cardiologist’s or the echocardiography reports, when available. The diagnosis of macrophage activation syndrome (MAS) was retained with the same methodology as SD. Hemophagocytosis on tissue sample (bone marrow, spleen, and lymph node) was systematically sought but not required for the diagnosis. HLH-2004 score and Hscore were calculated to assist in the classification of patients but were not mandatory to retain the diagnosis.

Two clinical patterns of SD were considered: 1) a systemic course defined by at least one systemic episode (with predominant fever, rash, lymphadenopathy, and/or serositis) lasting <6 months and followed by complete remission lasting >6 months; or 2) a chronic course defined by the persistence for >6 months of active disease with predominant joint involvement. Controlled disease was defined as clinically asymptomatic with normal acute phase reactants for >12 months. Recovery was defined as controlled disease, without any treatment, after >12 months of follow-up.

### 2.3. Statistical Analysis

AOSD and sJIA as well as disease course (chronic vs. systemic) were compared at the time of SD diagnosis. For non-normally distributed continuous variables descriptive analyses were presented as medians (interquartile range, IQR) and as means (±standard deviation, SD) for normally distributed variables. For qualitative variables, descriptive analyses were represented as frequencies and percentages of available data. Comparison between groups were performed using Mann–Whitney U-test, Student’s T test, Fisher’s exact test or chi-square test for non-parametric statistical adapted to the category of the variable and depending on whether the data were normally distributed or not. Before the multivariate analysis, missing values were imputed with multiple imputations with chained equations using the classification and regression trees algorithm. The multivariate analysis model was a generalized linear model where variables were selected as follows: collinearity was checked according to a correlation matrix and using variance inflation factor (VIF). A VIF under 5 was considered to reflect no or weak collinearity. The variable selection for multivariate analysis was performed by keeping all variables from a LASSO regression model with non-zero coefficients after L2 penalization. All the tests were two-tailed and a *p*-value of <0.05 was considered significant. Statistical analyses were performed using R version 4.1.2 (R foundation for Statistical Computing, Vienna, Austria). The packages used for the analysis were: compare Groups, glmnet and mice [20,21,22].

### 2.4. Ethics

The AURAL STILL study was approved by the institutional review board and the French CNIL (#19_348). The study is registered on ClinicalTrials.gov (#NCT05055882, accessed on 17 September 2022). All patients or legal representatives were sent written information about the study. In accordance with French laws on non-interventional retrospective studies, no written informed consent was required for inclusion.

## 3. Results

### 3.1. Characteristics of the Study Population

238 patients with SD (102 men and 136 women) were included in the study. Among them, 86 had sJIA and 152 had AOSD. The diagnosis was established in an internal medicine department in 77% of AOSD patients, in a rheumatology department in 14% and in other departments (infectious diseases, intensive care units, pediatrics) in 9%. For the whole population, the median age at SD diagnosis was 26.6 [10.5–43.4] years, 6.6 [2.8–12.3] years for sJIA and 38.3 [27.1–51.7] years for AOSD. Forty-one patients (17.2%; 27% of AOSD patients) were over 50 years at the onset of the disease. A history of autoimmune or autoinflammatory disease was found in 11 patients (five endocrinopathies, two neurological diseases, one axial spondyloarthritis, and three others).

At the time of SD diagnosis, leukocytosis (i.e., WBC ≥10 G/L) occurred in 81.2% of patient and the median WBC was 15.9 G/L (IQR, 11.8–20.3). Liver function tests were abnormal in 53.7% of patients and SF was increased in 95% (median, 2200 µg/L; IQR, 541–8000). GF was available in 125 patients, in whom it was <20% in 66.4% of cases. Quite surprisingly, RF was identified in 14/137 (10.2%) patients and ANA were positive in 42/191 (22.0%) patients. Anti-extractable nuclear antigens were found in only 2/159 (1.3%).

Of the 102 available bone marrow aspiration analyses, 14 (13.7%) revealed hemophagocytosis and two (1.9%) erythroblastopenia. The diagnosis of MAS was retained in 10/14 (71.4%). Of the 37 available bone marrow trephine biopsy reports, three (8.1%) clearly notified “MAS”, 20 (54.1%) “nonspecific inflammation”, 13 (35.1%) were “normal”, while a <10% lymphoid infiltration was found on the last one. Twenty-four patients underwent a lymph node biopsy; all revealed nonspecific or reactive inflammation.

### 3.2. Comparison between the Systemic and the Chronic Patterns

The disease course was systemic in 159 (66.8%) patients and chronic in 55 (23.1%) patients. Due to missing follow-up data, 24 (10.1%) patients could not be classified as systemic or chronic. A comparison of the epidemiological and clinical characteristics between the systemic and chronic patterns is presented in Table 1.

The sensitivity of the classification criteria did not differ between chronic and systemic phenotypes. The median diagnostic delay was significantly longer for chronic SD than for systemic SD (47 vs. 33 days; *p =* 0.018). Winter was the season when the disease occurred most often (32.5% of cases) and spring was the season when it occurred least (20.2%).

Among the symptoms at SD onset, skin rash, splenomegaly, and digestive involvement were significantly more frequent in patients with a systemic course.

Detailed comparisons of biological characteristics between the systemic and chronic patterns are presented in Table 2. Patients with systemic disease had significantly higher WBC and polymorphonuclear neutrophil (PMN) counts. These patients had significantly lower GF and were more likely to have GF < 20%. In contrast, SF levels were not different. Patients with systemic course had significantly higher levels of aspartate transaminase (AST), as well as LDH levels and CRP concentrations.

### 3.3. Comparison between sJIA and AOSD Patients

Among the 238 SD patients, 86 had sJIA and 152 had AOSD. In patients with sJIA, the ILAR criteria were met in 35/69 (50.7%) of patients, the PRINTO criteria in 40/72 (55.6%), at least one of two in 44/66 (66.7%), and both in 31/66 (46.9%). In patients with AOSD, 87/143 (60.8%) fulfilled the Yamaguchi criteria and 94/143 (65.7%) the Fautrel criteria. At least one of the two sets of criteria was met in 99/144 (68.8%) and both in 82/142 (57.3%) of cases. A comparison of the characteristics of sJIA and AOSD patients is shown in Table 3.

There were no statistically significant differences between sJIA and AOSD patients in terms of sex-ratio or geographical origin. There was no seasonality in children as in adults (respectively, *p =* 0.274 and *p =* 0.120). The clinical presentation was similar, except for sore throat (94.1 vs. 55.6%, *p* < 0.001) and myalgia (37.2 vs. 22.5%; *p* = 0.034), which were more frequent in the AOSD group. There were more differences in biological characteristics and particularly for the median SF concentration, which was 2.6 times higher in AOSD than in sJIA. For the other parameters the differences did not seem clinically relevant, except for ALT which were more often abnormal in AOSD and CRP which median was 1.55 times higher in AOSD.

### 3.4. Outcomes and Prognostic Factors

In 39 patients (16.4%), the initial manifestation of SD was a life-threatening complication. MAS was the most frequent. It complicated SD in 26 patients (10.9%) and was inaugural in 16 of them (61.5%). Most patients with MAS (*n* = 20) were reported in a previous publication by our team (in the setting of the AURAL STILL study) [23]. Other inaugural life-threatening complications included: 10 cases of myocarditis (4.2%), six cases of cardiac tamponade (2.5%), three cases of interstitial pneumonia (1.3%), two cases of pulmonary arterial hypertension (0.8%), and two cases of thrombotic microangiopathy (0.8%).

After a median follow-up of 44.6 [20.0–88.9] months, two patients developed lymphomas (58 and 82 months after SD diagnosis), one had a bone marrow failure (delay: 72 months) and one a clonal hypereosinophilia (delay: 21 months).

Three deaths (1.3%) were reported and two of them were related to SD. One death was secondary to thrombotic microangiopathy in a patient with AOSD, and the second was an in-hospital cardiac arrest, the exact origin of which was unspecified but related to sJIA (by the treating physician). The third death was due to non-Hodgkin lymphocytic lymphoma.

At last follow-up, 90/225 patients (40%) had achieved recovery, 82/225 (36.4%) controlled disease, 40/225 (17.8%) partially controlled disease, and 13/225 (5.8%) had a stable or worsening disease. Unsurprisingly, patients with a chronic pattern had less recovery and more incomplete remission (*p* < 0.001). There was no significant difference in outcome between sJIA and AOSD patients (Table 3).

In the unadjusted analysis, the first clinical events predictive of complete remission or recovery were fever (97.6 vs. 86.2%, *p =* 0.002) and skin rash (78.1 vs. 54.7%, *p =* 0.001). Initial joint involvement was not predictive of poor disease control (90 vs. 86.4%, *p =* 0.569). Regarding biological data, high LDH was a poor prognosis factor (median [IQR]; 373 (247–493) vs. 513 (280–720) U/L, *p =* 0.039). In contrast, WBC count, SF and GF levels were not associated with any particular outcome (Appendix A).

Regarding disease phenotypes, the adjusted analysis revealed that the presence, at the time of diagnosis, of splenomegaly (OR [95% CI]: 3.86 [1.14–18.07], *p* = 0.05), skin rash (OR [95% CI]: 2.34 [1.07–5.17], *p =* 0.03), high PMN count (OR [95% CI]: 1.09 [1.03–1.16] *p =* 0.007) and MAS (OR [95% CI]: 7.81 [1.43, 146.54], *p =* 0.05) were predictive of a systemic evolution (variables tested in the adjusted analysis: sex, MAS, skin rash, sore throat, lymphadenopathy, splenomegaly, digestive involvement, joint involvement, PMN count, serum ferritin level, antibiotic therapy prior to diagnosis; Appendix A).

## 4. Discussion

Since its initial description, Still’s disease has been a challenging nosological entity to position among other rheumatic diseases [24]. Initially, G.F. Still described it as a distinct disease from juvenile rheumatoid arthritis. In 1971, Bywaters described the disease in adults, with reference to that described in children by G.F. Still [25]. However, the later integration of SD within the large group of juvenile idiopathic arthritis has largely blurred the lines. The absence of specific signs, together with the intrinsic character of a diagnosis of exclusion, make it a disease diagnosed by default. The obscurity of its pathogenesis and its likely dynamic nature, mixing autoinflammation in the early stages and autoimmunity in the late/prolonged stages, further complicate its characterization. Fortunately, the advent of biotherapies has changed the understanding of the disease and its natural history. At present, in the absence of robust prospective data, experts are seeking to rely on meta-analyses of retrospective series to try to perfectly define the contours of a pathological spectrum that is still somewhat vague.

In this large retrospective multicenter study, we analyzed data from patients with a diagnosis of sJIA and with a diagnosis of AOSD. Such a methodology was mandatory to compare one with the other. We also combined data from adults and children in order to maximize the chances of identifying prognostic factors. Previous studies have already shown that, epidemiologically, clinically and biologically, sJIA and AOSD are part of the continuum of Still’s disease [8,9,26]. The results of our study confirm that there are only a few differences between sJIA and AOSD, which can probably be explained by the age at disease onset rather than by different pathophysiological mechanisms [17,27].

The epidemiology and manifestations of SD may vary according to geographical origin and ethnicity [2,4,6,28,29,30] (Table 4). Combining sJIA and AOSD, the median age at disease onset in our cohort was 26.6 years and cannot be compared with previous sJIA or AOSD series. Considering only patients with AOSD, the median age at disease onset was 38.3 years and was similar to the ages reported in the previous Italian and French cohorts [4,28] (Table 4).

The systemic pattern was the most frequent (about two-third of the cohort). However, except for a shorter diagnostic delay, the higher frequencies of skin rash and splenomegaly, we did not identify significant differences in the clinical presentation between systemic and chronic SD patterns. In contrast, some biological features may help distinguish the systemic form (i.e., higher PMN count, higher CRP and higher LDH levels). A decrease in GF fraction is also indicative of a subsequent systemic course. Thus the dichotomic view of the disease still needs to be confirmed.

The distribution of disease phenotypes was not different between sJIA and AOSD. This may explain and illustrate why previous reports have yielded conflicting results [9,31]. It can be hypothesized that this divergence may result from changes in (children’s) disease management.

To evaluate the performance of the selection criteria, to avoid confirmation bias and to reflect real-life practice, cases were included on the basis of the clinician’s diagnosis (further reviewed and confirmed by two independent investigators). The classification criteria were met in 66.7% of children and 68.8% of adults. In addition, and rather surprisingly ANA were positive in 22% of the patients, which was higher than previously reported (Table 4) and higher than expected in the general population. However, most of these studies have as inclusion criteria a diagnosis depending on the Yamaguchi criteria (one of the minor criteria of which is the absence of ANA). In their cohort Asanuma et al. reported a frequency of positivity of ANA of 25.8% while the diagnosis depended on the clinician’s judgment [5]. Moreover, in our study, ANA had no specificity and only two patients had anti-ENA antibodies, without any manifestation suggestive of a definite autoimmune disease.

MAS was the most common life-threatening complication of SD. In the literature, the prevalence of MAS is highly variable ranging from 3% to 22% in a recent Japanese study [4,6]. In our cohort, 11% of patients underwent MAS and none died. The detection and prediction of MAS during SD have been analyzed in depth previously [23,32,33]. Overall, during the study period (i.e., over twenty years), three deaths were recorded and two of them were related to SD. This is consistent with the generally good prognosis of SD, although recent Asian series have reported mortality rates around 10% [34,35]. These series are not consistent with the rest of the literature and only future studies will tell us whether there is a change in the epidemiology of the disease, local or global, or whether these studies had undetected artefacts.

Few studies have investigated the prognostic factors in SD [28,34,35,36,37]. We found that fever and rash at diagnosis were good prognostic factors. This is consistent with the previous report by Gerfaud-Valentin et al. showing high fever is predictive of monocyclic evolution [28]. In contrast, high LDH levels were associated with a poor prognosis. We were not able to detect any predictive value for ferritin levels, arthralgia/arthritis, or thrombocytopenia, but splenomegaly and higher PMN count were associated with progression to a systemic rather than a chronic articular pattern [3,28,37,38,39].

Our study has some limitations. First, as it was a retrospective study, a number of data were missing. Second, our selection criteria (i.e. diagnosis based on that of the treating clinician) may have made our population more heterogeneous. However, comparison with the literature revealed broadly similar characteristics. Third, due to the long duration of the study period, the management of SD has evolved and changed the natural history of the disease.

## 5. Conclusions

In this large retrospective study, we found no major differences between sJIA and AOSD. Furthermore, the dichotomy between systemic and chronic patterns is probably a convenient way to categorize the disease at a given time or for research purposes, but it is highly uncertain whether there is an initial determinism. High fever and rash at diagnosis are associated with a monocyclic systemic course, but rather because they prompt earlier and more aggressive treatment. These patterns have to be confirmed in further studies.

## Figures and Tables

**Table 1 jcm-11-06703-t001:** Epidemiological and clinical characteristics according to the disease phenotype.

Characteristics		Chronic (*n* = 55)	Systemic (*n* = 159)	Total (*n* = 238)	*p*-Values (Univariate) *
**Epidemiology**					
	Men/Women	19/36	74/85	102/136	0.165
	Median age at diagnosis, y [IQR]	26.4 (11.0–48.1)	26.4 (11.6–39.3)	26.6 (10.5–43.4)	0.827
	Median diagnostic delay, d [IQR]	47 (30–200)	30 (18–69)	33 (19–90)	**0.018**
	Caucasian origin, N (%)	29/41 (70.7%)	87/128 (68%)	132/186 (71.0%)	0.890
	sJIA, N (%)	22 (40.0%)	56 (35.2%)	86/238 (36.1%)	0.637
	AOSD, N (%)	33 (60.0%)	103 (64.8%)	152/238 (63.9%)	0.637
**Classification**					
**criteria**	ILAR, N (%)	7/15 (46.7%)	28/51 (54.9%)	35/69 (50.7%)	0.7189
	PRINTO, N (%)	7/13 (53.8%)	33/51 (64.7%)	40/72 (55.6%)	0.530
	Yamaguchi, N (%)	16/28 (57.1%)	65/99 (65.7%)	87/143 (60.8%)	0.545
	Fautrel, N (%)	15/28 (53.6%)	73/99 (73.7%)	94/143 (65.7%)	0.070
**Clinical features**					
	Fever, N (%)	50/55 (90.9%)	150/156 (96.2%)	222/235 (94.5%)	0.159
	Fever > 39 °C, N (%)	29/37 (78.4%)	122/136 (89.7%)	168/193 (87.0%)	0.092
	Joint involvement N (%)	51/55 (92.7%)	138/157 (87.9%)	210/236 (89.0%)	0.460
	Skin rash, N (%)	29/53 (54.7%)	123/156 (78.8%)	167/233 (71.7%)	**0.001**
	Sore throat, N (%)	23/52 (44.2%)	84/150 (56.0%)	117/225 (52.0%)	0.192
	Myalgia, N (%)	12/49 (24.5%)	54/152 (35.5%)	72/225 (32.0%)	0.209
	Lymphadenopathy, N (%)	11/35 (31.4%)	59/125 (47.2%)	81/179 (45.3%)	0.142
	Splenomegaly, N (%)	1/43 (2.3%)	24/129 (18.6%)	29/192 (15.1%)	**0.018**
	Hepatomegaly, N (%)	3/43 (7.0%)	19/126 (15.1%)	27/188 (14.4%)	0.271
	Pericarditis, N (%)	8/52 (15.4%)	30/158 (19.0%)	41/233 (17.6%)	0.706
	Pleurisy, N (%)	5/55 (9.1%)	26/159 (16.4%)	32/238 (13.4%)	0.273
	Digestive involvement, N (%)	4/52 (7.7%)	34/154 (22.1%)	42/229 (18.3%)	**0.035**
	Weight loss, N (%)	11/32 (34.4%)	30/107 (28.0%)	52/157 (33.1%)	0.639

* Statistically significant *p*-values are in bold font. AOSD: adult-onset Still’s disease; d: days; ILAR: International League of Associations for Rheumatology; PRINTO: Pediatric Rheumatology International Trials Organization; sJIA: systemic juvenile idiopathic arthritis; y: years.

**Table 2 jcm-11-06703-t002:** Biological characteristics according to the disease phenotype.

Biological Characteristics	Chronic (*n* = 55)	Systemic (*n* = 159)	Total (*n* = 238)	*p*-Values (Univariate) *
WBCs, median [IQR], G/L	14.0 (8.8–17.3)	16.0 (12.8–20.4)	15.9 (11.8–20.3)	**0.020**
PMNs, median [IQR], G/L	10.0 (6.0–13.2)	13.0 (9.9–17.0)	12.5 (8.6–16.3)	**<0.001**
Hemoglobin, mean (SD), g/L	117 (±18.3)	113 (±17.5)	113 (±17.3)	0.291
Platelets, median [IQR], G/L	376 (300–526)	374 (265–492)	375 (267–492)	0.386
Serum ferritin, median [IQR], µg/L	1976 (242–5000)	2450 (598–8616)	2200 (541–8000)	0.089
Ferritin glycosylated fraction, median [IQR], %	19 (14–28)	14.0 (9–22)	15 (10–23)	**0.041**
Ferritin glycosylated fraction <20%, N (%)	13/26 (50.0%)	60/84 (71.4%)	83/125 (66.4%)	**0.043**
CRP, median [IQR], mg/L	131 (74–210)	177 (104–252)	164 (97–245)	**0.029**
AST, median [IQR], U/L	31 (22–48)	42(27–86)	39 (26–80)	**0.025**
ALT, median [IQR], U/L	24 (15–40)	33 (14–87)	30 (14–76)	0.227
GGT, median [IQR], U/L	40 (21–94)	57 (23–161)	55 (22–152)	0.163
LDH, median [IQR], U/L	298 (216–397)	392 (272–570)	381 (266–572)	**0.020**
PT, median [IQR], %	81 (75–93)	76 (64–88)	78 (68–88)	0.075
Fibrinogen, mean (SD), g/L	6.1 (±2.4)	6.4 (±2.2)	6.2 (±2.2)	0.519
Triglycerides, median [IQR], mmol/L	1.54 (1.11–1.88)	1.55 (1.16–2.21)	1.55 (1.17–2.02)	0.614
Rheumatoid factor, N (%)	3/33 (9.1%)	7/93 (7.5%)	14/137 (10.2%)	0.721
ANA, N (%)	15/44 (34.1%)	24/129 (18.6%)	42/191 (22.0%)	0.056

* Statistically significant *p*-values are in bold font. ANA: antinuclear autoantibody; ALT: alanine transaminase AST: aspartate transaminase; CRP: C-reactive protein; GGT: gamma-glutamyltransferase; LDH: lactate dehydrogenase; PMNs: polymorphonuclear neutrophils; PT: prothrombin time; WBCs: white blood cells.

**Table 3 jcm-11-06703-t003:** Epidemiological, clinical and laboratory features of sJIA and AOSD patients.

Characteristics		sJIA (*n* = 86)	AOSD (*n* = 152)	*p*-Values (Univariate) *
**Epidemiology**				
	Men/Women	38/48	64/88	0.861
	Median age at diagnosis, y [IQR]	6.6 (2.8–12.3)	38.3 (27.1–51.7)	
	Caucasian origin, N (%)	46/66 (69.7%)	86/120 (71.7%)	0.909
**Evolution**				
	Chronic, N (%)	22 (25.6%)	33 (21.7%)	0.637
	Systemic, N (%)	56 (65.1%)	103 (67.8%)	0.637
	Unclassified course, N (%)	8 (9.3%)	16 (10.5%)	0.637
**Clinical features**				
	Fever, N (%)	81/86 (94.2%)	141/149 (94.6%)	1.000
	Joint involvement N (%)	75/86 (87.2%)	135/150 (90.0%)	0.658
	Skin rash, N (%)	60/84 (71.4%)	107/149 (71.8%)	1.000
	Sore throat, N (%)	20/82 (24.4%)	97/143 (67.8%)	**<0.001**
	Myalgia, N (%)	18/80 (22.5%)	54/145 (37.2%)	**0.034**
	Lymphadenopathy, N (%)	29/61 (47.5%)	52/118 (44.1%)	0.776
	Splenomegaly, N (%)	12/69 (17.4%)	17/123 (13.8%)	0.651
	Hepatomegaly, N (%)	13/69 (18.8%)	14/119 (11.8%)	0.264
	Pericarditis, N (%)	17/84 (20.2%)	24/149 (16.1%)	0.538
	Pleurisy, N (%)	11/86 (6.98%)	21/152 (13.8%)	0.980
	Digestive involvement, N (%)	17/84 (20.2%)	25/145 (17.2%)	0.698
	Weight loss, N (%)	13/48 (27.1%)	39/109 (35.8%)	0.377
**Biological**				
**characteristics**	WBCs, median [IQR], G/L	17.4 (14.4–22.4)	15.0 (10.9–18.9)	**0.001**
	PMNs, median [IQR], G/L	13.4 (9.6–17.9)	12.0 (8.5–15.3)	0.094
	Serum ferritin, median [IQR], µg/L	1076 (300–3077)	2821 (990–10,224)	**<0.001**
	CRP, median [IQR], mg/L	120 (82–194)	186 (105–268)	**<0.001**
	AST, median [IQR], U/L	33 (24–45)	47 (27–96)	**0.004**
	ALT, median [IQR], U/L	14 (9–27)	50 (21–106)	**<0.001**
	LDH, median [IQR], U/L	391 (298–505)	378 (242–588)	0.767
	PT, median [IQR], %	71 (58–84)	79 (69–90)	**0.011**
	Fibrinogen, mean (SD), g/L	5.7 (±1.9)	6.6 (±2.3)	**0.009**
	Rheumatoid factor, N (%)	2/32 (6.25%)	12/105 (11.4%)	0.520
	ANA, N (%)	13/59 (22.4%)	29/133 (21.8%)	1.000

* Statistically significant *p*-values are in bold font. ANA: antinuclear autoantibody; ALT: alanine transaminase; AOSD: adult-onset Still’s disease; AST: aspartate transaminase; CRP: C-reactive protein; GGT: gamma-glutamyltransferase; LDH: lactate dehydrogenase; PMNs: polymorphonuclear neutrophils; PT: prothrombin time; sJIA: systemic juvenile idiopathic arthritis; WBCs: white blood cells; y: years.

**Table 4 jcm-11-06703-t004:** Comparison of our cohort with five previously published series.

	Present Study	Pouchot 1991 [2]	Pay 2006 [31]	Gerfaut-Valentin 2014 [28]	Sfriso 2016 [4]	Hu 2019 [29]
Country	France	Canada	Turkey	France	Italy	China
Case number	238	62	120	57	245	517
Diagnosis criteria	CO	Composite	Y (AOSD) D (SJIA)	Y or F	Y	Y
Age at onset, y	26.6 *	24 *	NA	36*	38.8 §	37.7 §
Women	57%	45%	53%	53%	47%	72%
Fever	95%	100%	94%	95%	93%	100%
Arthralgia	89%	100%	95%	95%	93%	73%
Sore throat	52%	92%	58%	53%	62%	61%
Skin rash	72%	87%	78%	77%	68%	80%
Lymphadenopathy	45%	74%	40%	60%	60%	51%
Splenomegaly	15%	55%	42%	30%	NA	34%
Hepatomegaly	14%	41%	45%	21%	42%	7%
Pericarditis	18%	27%	10%	19%	17%	14%
WBCs ≥10,000/mm^3^	81%	94%	NA	72%	81%	86%
PMNs ≥ 80%	58%	88%	71%	77%	70%	78%
Elevated liver enzymes	54%	76%	55%	66%	54%	62%
SF level						
>1 × normal	95%	NA	89%	82%	80%	96%
>5 × normal	73%	NA	60%	NA	NA	79%
GF < 20%	66%	NA	NA	76%	NA	NA
Positive ANA	22%	11%	NA	8%	10%	9%
Positive RF	10%	6%	NA	0%	4%	6%

* Median § Average. ANA: antinuclear autoantibody; CO: clinician’s opinion; D: Durban criteria; F: Fautrel criteria; PMNs: polymorphonuclear neutrophils; RF: rheumatoid factor; SF: serum ferritin; WBCs: white blood cells; y: years; Y: Yamaguchi Criteria.

## Data Availability

Additional data may be obtained on reasonable request from the corresponding author.

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
