# Peer review of "The Spectrum of Still’s Disease: A Comparative Analysis of Phenotypic Forms in a Cohort of 238 Patients"

_jcm, 2022, doi:10.3390/jcm11226703_

Round 1
Reviewer 1 Report
In this study, authors sought to assess the heterogeneity of the SD spectrum. They analyzed a multicenter cohort of patients with Still's disease with high number of patients included. The topic and effort taken is of great importance in the field.
However I have some concerns I would like to clarify
- Patients were followed between Jan 2001 and Jun 2022. I would like to ask for time of follow up and its comparison between SJIA and AOSD patients
- ANA with titer 1:160 were considered positive by what test, IF?
-chronic course was defined > 6 mo of active disease with predominant joint involvement. What does predominant join involvement mean? Destructive arthritis? Was a scenario with chronic activity without predominant joint involvement possible?
Were there clinicaly asymptomatic and laboratory active patients observed? Or clinically active and laboratory asymptomatic?
line 158: . Forty-one patients (17.2%) were over 50 years at the onset of the disease. It is almost 27% of AOSD - I would add information if you agree
Lines 168-169 Of the 102 available bone marrow aspiration analyses, 14 (13.7%) revealed hemophagocytosis and two (1.9%) erythroblastopenia. The diagnosis of MAS was retained in 10/14 (71.4%). Of the 37 available bone marrow biopsy reports.
This is unclear 37 available reports 102 available analyses. Are biopsy reports refered to trepanobiopsy?
Discussion
Now is not easy to follow and to understand main results. First paragraph is too long and more suitable for introduction
I suggest to rewrite discussion according to the check list of the aims:
The aim of the study was: 1) to re- 82 evaluate the dichotomous view of the disease; 2) to compare sJIA and AOSD; and 3) to 83 look for predictive factors for progression and outcome.
Tekst should be carefully analysed. In ex. in line 314: This is consistent with the mixed results from previous reports . It cannot be consistent with mix results. Mix results mean that some are in agreement other contrary. Pleas explain in more details
Conclusion
Conclusion that: These findings seem to support the current “biphasic model” with an early therapeutic window of opportunity, and its counterpart: the treat- to-target strategy. In my opinion is not supported by the results.
As authors stated earlier this patterns have to be confirmed in further studies
Moreover data about treatment are not given in the study, window opportunity can not be analysed. This should be omitted.
Author Response
Thank you for your review. Here are some answers to the points you identified. Please see the attachment.
1 « Patients were followed between Jan 2001 and Jun 2022. I would like to ask for time of follow up and its comparison between SJIA and AOSD patients”
Time of follow-up is mentioned L241 "After a median follow-up of 44.6 [20.0-88.9] months". Comparison between SJIA/AOSD were not mentioned because similar (47.3 vs 43.1).
2 “ANA with title 1:160 were considered positive by what test, IF?”
Indirect immunofluorescence, indeed.
Modification L110: “ANA were searched by indirect immunofluorescence, with titers ≥160 were considered positive”
3“-chronic course was defined > 6 mo of active disease with predominant joint involvement. What does predominant join involvement mean? Destructive arthritis? Was a scenario with chronic activity without predominant joint involvement possible?”
These patients presented with inflammatory arthralgia, arthritis or destructive arthritis with little or no systemic involvement. A phenotype “chronic articular/polycyclic systemic" (as Cush's classification (1)) was possible in some patients and accounts for part of the "unclassifiable" phenotypes.
1) Cush, J.J.; Medsger, T.A.; Christy, W.C.; Herbert, D.C.; Cooperstein, L.A. Adult-Onset Still’s Disease. Clinical Course and Outcome. Arthritis Rheum. 1987, 30, 186–194, doi:10.1002/art.1780300209.
4“Were there clinically asymptomatic and laboratory active patients observed? Or clinically active and laboratory asymptomatic?”
No, both were related for all of our population
5 “line 158: . Forty-one patients (17.2%) were over 50 years at the onset of the disease. It is almost 27% of AOSD - I would add information if you agree”
Modification L159: “Forty-one patients (17.2%; 27% of AOSD patients) were over 50 years”
6 “Lines 168-169 Of the 102 available bone marrow aspiration analyses, 14 (13.7%) revealed hemophagocytosis and two (1.9%) erythroblastopenia. The diagnosis of SAM was retained in 10/14 (71.4%). Of the 37 available bone marrow biopsy reports. This is unclear 37 available reports 102 available analyses. Are biopsy reports referred to trepanobiopsy?”
There was:
- 102 bone marrow aspiration
- 37 bone marrow trepanobiopsy
Modification L171: “Of the 37 available bone marrow trephine biopsy reports”
7 “Now is not easy to follow and to understand main results. First paragraph is too long and more suitable for introduction”
The authors have chosen to keep the introduction short (as also requested in the publisher's guidelines), so that the reader can quickly get to the core hypothesis and findings of their study. However, the problematic needs to be explained in more detail and put into perspective, which explains why the first paragraph of the discussion may seem rather long.
8 “I suggest to rewrite discussion according to the check list of the aims:The aim of the study was: 1) to re-82 evaluate the dichotomous view of the disease; 2) to compare sJIA and AOSD; and 3) to 83 look for predictive factors for progression and outcome.”
The authors have preferred to keep the frame of the discussion.
9 “Tekst should be carefully analysed. In eg. in line 314: This is consistent with the mixed results from previous reports . It cannot be consistent with mix results. Mix results mean that some are in agreement other contrary. Please explain in more details”
The literature shows contradictory results in the distribution of phenotypes between SJIA and AOSD. The authors meant that the results of this study (ie distribution of disease phenotypes not different between sJIA and AOSD) were consistent with the heterogeneous results of previously published studies.
This sentence (L317) has been corrected to "this may explain and illustrate why previous reports have yielded conflicting results"
10 “Conclusion that: These findings seem to support the current “biphasic model” with an early therapeutic window of opportunity, and its counterpart: the treat-to-target strategy. In my opinion is not supported by the results. As authors stated earlier this patterns have to be confirmed in further studiesMoreover data about treatment are not given in the study, window opportunity can not be analysed. This should be omitted.”
The last sentence (L361) has been modified: “these patterns have to be confirmed in further studies.”
Reviewer 2 Report
In this multicentre retrospective analysis, the authors compare patients with systemic juvenile arthritis with adult patients having AOSD. Fever and skin rash were predictive of complete remission, which makes sense, since those symptoms are leading to prompt treatment. LDH, on the other side, was prognostic for a poor outcome.
To overcome the bias of the retrospective design, the study group has chosen to review the records prospectively and validate the diagnosis, which I think it is the optimal strategy.
I found interesting the fact that 102 of 238 patients underwent a bone marrow biopsy, a very high number. The results section was, overall, very well written without unnecessary information flow.
Only about 60% of the patients fulfilled the classification criteria, which underlines the need for new diagnostic tools. Table 4 shows how important such multicentre studies are. This French article is as valuable for the field of rare diseases like AOSD as older cohort studies from Italy and China.
I recommend accept with very small revision:
Abstract, line 46: period is missing “…group. Fever…”
Author Response
Thank you very much for your review. The sentence, L46, has been corrected.
Reviewer 3 Report
In this report, the authors compared the clinical features of patients with systemic course and chronic course in patients with Still’s disease, compared the clinical features among patients with juvenile and adult onset cases, as well as identifying predictive factors for progression and outcomes. The number of the patients in the study was quite large. However, there are several concerns.
1. L70. According to Cush classification (ref 12), AOSD has 4 subtypes not 3 subtypes, please check.
2. L110. Why did the authors choose ANA at 1:160 as a positive test, not at 1:80 as generally accepted? Please clarify
3. L111. Why only pleurisy/pleuritis was assessed by imaging only? Pleurisy/pleuritis can be obtained by typical characteristic of the symptoms and audible pleural friction rubs.
4. L112. Was pericarditis confirmed by cardiologist refers to “pericardial friction rubs”?
5. Table 1 and 2. The authors compared the clinical features and laboratory findings among the systemic course and the chronic course patients. These combined the adults and pediatric patients into one group. It might be better if the authors could separate the patients in to systemic course (AOSD vs. sJIA), and chronic course (AOSD vs. sJIA), as these could make better vision of the difference in clinical course between the two age groups. In doing this, not only a clear picture between the AOSD and sJIA, table 3 is not necessary (as it will be covered in the new table 1 and 2).
6. The authors mentioned in the aims of the study that the predictive factors for progression and outcome. Unfortunately, the authors did not verify the progression and outcomes clearly. It might be better if the authors could showed in a table for variables that were used in the analysis for potential factors for progression and outcomes.
7. Table 4. Interestingly, the 22% ANA positive in this study was rather high when compared with previously published data (despite the authors used the cut-off titer of 1:160). Please explain this finding.
Author Response
Thank you for your review. Here are some answers to the points you identified. Please see the attachment.
1 According to Cush classification (ref 12), AOSD has 4 subtypes not 3 subtypes, please check.
Indeed, this reference illustrated the prevalence of the different subtypes for an earlier version of the manuscript.
The classification into 3 phenotypes (monocyclic, polycyclic and chronic) is classically more used (1; 2).
Modification: We replace this ref by another more adapted (3)
1) Feist E, Mitrovic S, Fautrel B. Mechanisms, biomarkers and targets for adult-onset Still's disease. Nat Rev Rheumatol. 2018 Oct;14(10):603-618. doi: 10.1038/s41584-018-0081-x. PMID: 30218025; PMCID: PMC7097309.
2) Efthimiou P, Kontzias A, Hur P, Rodha K, Ramakrishna GS, Nakasato P. Adult-onset Still's disease in focus: Clinical manifestations, diagnosis, treatment, and unmet needs in the era of targeted therapies. Semin Arthritis Rheum. 2021 Aug;51(4):858-874. doi: 10.1016/j.semarthrit.2021.06.004. Epub 2021 Jun 13. PMID: 34175791.
3) Magadur-Joly, G.; Billaud, E.; Barrier, J.H.; Pennec, Y.L.; Masson, C.; Renou, P.; Prost, A. Epidemiology of Adult Still's Dis-ease: Estimate of the Incidence by a Retrospective Study in West France. Ann Rheum Dis 1995, 54, 587–590, doi:10.1136/ard.54.7.587
2 Why did the authors choose ANA at 1:160 as a positive test, not at 1:80 as generally accepted? Please clarify
The positivity threshold reported by the laboratories of the participating centers is 1/160 (ANA 1/80 is found in 10 – 15% of healthy people while ANA 1:160 is found in 5% of healthy people).
3 Why only pleurisy/pleuritis was assessed by imaging only? Pleurisy/pleuritis can be obtained by typical characteristic of the symptoms and audible pleural friction rubs.
The presence of pleuritis/pleurisy was retained based on the opinion of the patient's referring clinician in the reports (including clinical symptoms in addition to imaging). In fact, a large majority of patients with pleuritis/pleurisy have had an X-ray or a CT scan. To compensate for missing data, we carried out a double check by the investigator on the available imaging.
Modification L111: Pleuritis/pleurisy was assessed by a double check of the investigator on medical imaging when available, in addition to medical reports
4 Was pericarditis confirmed by cardiologist refers to “pericardial friction rubs”?
Pericarditis was assessed based on reports
1) echocardiography
2) Cardiological opinion based on clinical/electrocardiogram/ amnamnesis if echocardiography was normal
5 Table 1 and 2. The authors compared the clinical features and laboratory findings among the systemic course and the chronic course patients. These combined the adults and pediatric patients into one group. It might be better if the authors could separate the patients in to systemic course (AOSD vs. sJIA), and chronic course (AOSD vs. sJIA), as these could make better vision of the difference in clinical course between the two age groups. In doing this, not only a clear picture between the AOSD and sJIA, table 3 is not necessary (as it will be covered in the new table 1 and 2).
In our opinion, the literature and our results point in the direction of the same disease (or at least the same continuum) between SJIA and AOSD. There would therefore be no reason to dichotomize the systemic/chronic analysis in relation to the pediatric character of the disease or not.
Modification: We add the following table (L488) in Supplementary material
6 The authors mentioned in the aims of the study that the predictive factors for progression and outcome. Unfortunately, the authors did not verify the progression and outcomes clearly. It might be better if the authors could showed in a table for variables that were used in the analysis for potential factors for progression and outcomes.
The co-variables of the adjusted analysis (evolution towards a particular phenotype) have been added (L264).
7 Table 4. Interestingly, the 22% ANA positive in this study was rather high when compared with previously published data (despite the authors used the cut-off titer of 1:160). Please explain this finding.
Most of these studies have as inclusion criteria a diagnosis depending on the Yamaguchi criteria (one of the minor criteria of which is the absence of ANA). One of rare study with a inclusion criteria based on physician’s judgment, as our, found positive ANA in 26% of patients (1).
1) Asanuma, Y.F.; Mimura, T.; Tsuboi, H.; Noma, H.; Miyoshi, F.; Yamamoto, K.; Sumida, T. Nationwide Epidemiological Survey of 169 Patients with Adult Still’s Disease in Japan. Mod. Rheumatol. 2015, 25, 393–400, doi:10.3109/14397595.2014.974881.
Modification L324: “ANA were positive in 22% of the patients, which was higher than previously reported (Table 4) and higher than expected in the general population. However, most of these studies have as inclusion criteria a diagnosis depending on the Yamaguchi criteria (one of the minor criteria of which is the absence of ANA). In their cohort Asanuma et al. reported a frequency of positivity of ANA of 25.8% while the diagnosis depended on the clinician's judgment [5]. Moreover, in our study, ANA had no specificity and only two patients had anti-ENA antibodies, without any manifestation suggestive of a definite autoimmune disease.”
Round 2
Reviewer 3 Report
Although the authors had reponsed to the queries, a table that consited of univarialble and multivariable logistic regression analysis is need (for the predictive factors of systemic or chronic arthritis subtypes). This would make it more clear for the readers.
Author Response
Both tables have been added to the supplementary materials.
Please see the attachment.
